# A Review on Bitumen Aging and Rejuvenation Chemistry: Processes, Materials and Analyses

**Emiliano Prosperi** [1,*] and **Edoardo Bocci** [2]

1   Department of Construction, Civil Engineering and Architecture, Marche Polytechnic University, 60131 Ancona, Italy
2   Faculty of Engineering, eCampus University, 22060 Novedrate, Italy; edoardo.bocci@uniecampus.it
*   Correspondence: e.prosperi@pm.univpm.it

**Abstract:** During the last decades, extensive research has been carried out on using reclaimed asphalt pavement (RAP) material in the production of hot recycled mix asphalt. Unfortunately, the aged, stiff, and brittle binder in the RAP typically increases the mixture stiffness and can therefore cause fatigue and low-temperature damages. In the scientific literature, there are many studies concerning the aging and rejuvenation of bitumen, but there is a lack of up-to-date reviews that bring them together, especially those facing the phenomena from a chemical point of view. In this paper, a recap of the chemical aspects of virgin, aged, and rejuvenated bitumen is proposed in order to provide a useful summary of the state of the art, with the aim of both encouraging the use of an increasing quantity of RAP in hot mix asphalt and trying to give indications for further research.

**Keywords:** bitumen; aging; rejuvenation; reclaimed asphalt; recycling

## 1. Introduction

Nowadays, the world is living through the most severe environmental crisis ever, due to biodiversity loss; air, soil, and water pollution; resource exhaustion; and disproportionate land use. To face these issues, the concepts of circular economy and sustainability are getting more and more attention by administrations, technicians, and researchers. Indeed, these topics have become important themes of research with a steep increase in the number of papers and journals [1]. One of the sectors that greatly contributes to the environmental impact is road construction, which entails the exploitation of raw materials, the emission of pollutants, and the generation of wastes [2].

In the sector of road pavement engineering, the easiest way to promote the circular economy is to encourage the use of Reclaimed Asphalt (RAP). RAP is defined as removed and/or reprocessed pavement material composed of bitumen and aggregates. Four methods—hot in-plant recycling, hot in-place recycling, cold in-plant recycling, and cold in-place recycling—allow RAP recycling in new bituminous mixtures, determining the saving of virgin resources and the progressive cyclic reuse of this waste material, in a sustainable and circularly economic way [3].

The most widespread method to recycle RAP is the hot in-plant technique, which allows both the aggregates and binder contained in the RAP to be exploited. Unfortunately, the aged, stiff, and brittle binder from the RAP increases the mixture stiffness and can therefore cause fatigue and low-temperature damages [4,5]. For this reason, road agencies usually limit the maximum amount of RAP that can be recycled in new hot mixes [6]. With the aim of solving this issue, many products worldwide have been used with the function of rejuvenating agents, which allow restoring (fully or partly) the mechanical properties that the RAP binder loses with aging [7].

In the scientific literature, there are several studies concerning the aging and rejuvenation of bitumen. In particular, most of the studies are focused on how "improved" (in terms of performance or even environmental friendliness) bituminous binders behave

with aging and when blended with aged bitumen from RAP. For instance, the use of polymer-modified bitumen is one of the hottest topics in this field [8]. Bitumen including elastomeric or plastomeric polymers experiences different changes with aging with respect to neat bitumen, as a function of the polymer type and content [9–13]. Moreover, it is a theme of research whether RAP including polymer-modified bitumen is as recyclable as the RAP containing neat bitumen, in terms of final mix performance and emission during mix production [14,15]. Another technique to increase the bitumen performance, which is actually investigated in terms of aging behavior, deals with the use of nanomaterials as modifiers, such as fumed silica, clay, diatomite, titanium dioxide, graphene, and carbon nanotubes [16–22]. However, there are still some gaps of knowledge on the understanding of neat-bitumen-aging phenomena, particularly from a chemical point of view. Many researchers have been trying to fill these gaps in the last five years. As shown in Table 1 and Figure 1, the search for the keywords "bitumen" (or "asphalt", in the American dictionary) and "aging" or "rejuvenation" in the Scopus database has provided several occurrences when associated with chemical analyses. More than half of these publications come from Chinese authors (first author), but many studies have been carried out also in North America (the United States and Canada) and Europe.

**Table 1.** Number of papers indexed by Scopus in the period 2016–2021 on the theme of bitumen aging/rejuvenation chemistry (date of the research 27 May 2021).

| Keywords | Papers Published Since 2016 |
| --- | --- |
| "Bitumen/Asphalt", "Aging", "AFM" | 114 |
| "Bitumen/Asphalt", "Aging", "Chemistry" | 71 |
| "Bitumen/Asphalt", "Aging", "Chromatography" | 115 |
| "Bitumen/Asphalt", "Aging", "FTIR" | 395 |
| Total number of papers of bitumen aging chemistry | 589 |
| "Bitumen/Asphalt", "Rejuvenation", "AFM" | 24 |
| "Bitumen/Asphalt", "Rejuvenation", "Chemistry" | 15 |
| "Bitumen/Asphalt", "Rejuvenation", "Chromatography" | 29 |
| "Bitumen/Asphalt", "Rejuvenation", "FTIR" | 73 |
| Total number of papers on bitumen rejuvenation chemistry | 121 |

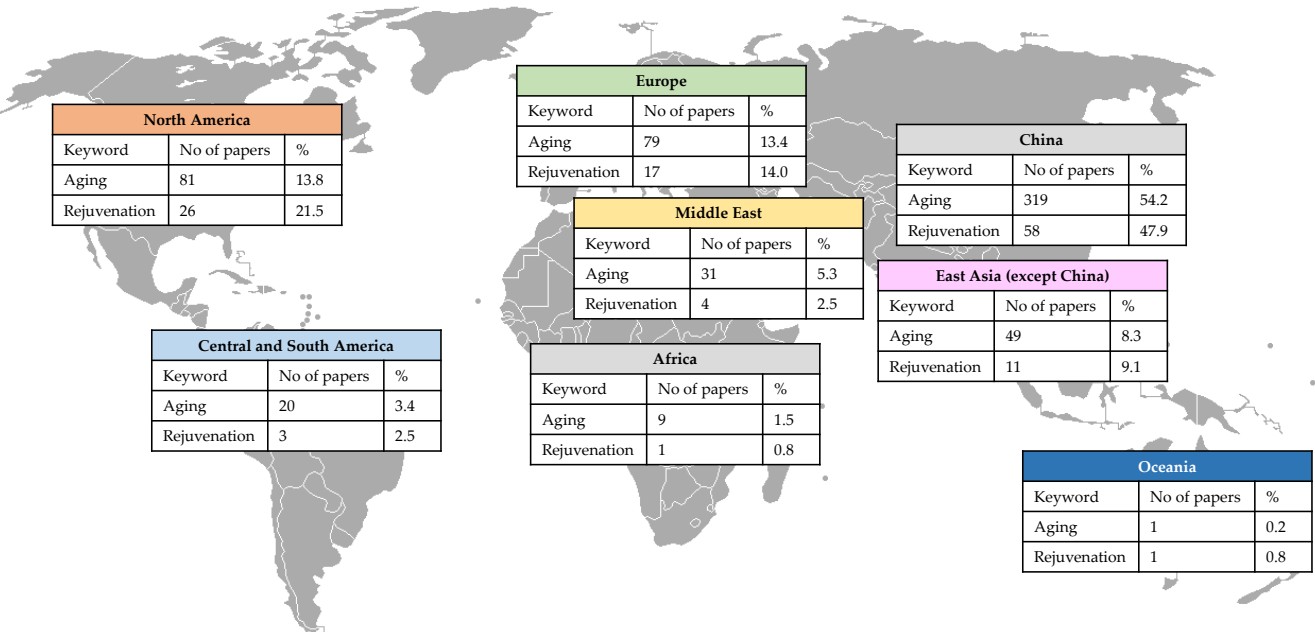

**Figure 1.** Origin of the papers indexed by Scopus in the period 2016–2021 on the theme of bitumen aging/rejuvenation chemistry.

This paper tries to recap the most recent research and innovation in the field of chemical analyses on virgin, aged, and rejuvenated bitumen. With respect to other reviews on aging and rejuvenation chemistry, the present state of the art is more updated (the most recent paper was published in 2009 [23]) and includes discussion about both processes (while the review by Loise et al. [24] only deals with rejuvenation, and specifically on additives). The aim is to provide a useful summary and some inputs for further research on this topic, in order to encourage the use of an increasing quantity of RAP in Hot Mix Asphalt (HMA).

## 2. Bitumen Chemistry

Bitumen is the visco-elasto-plastic material obtained through crude oil distillation. Basically, during this process, the various phases of the crude oil are separated due to the differences in their boiling and condensing temperatures [25]. A typical distillation process involves a first step in which the lighter components are separated, subjecting the crude oil to a temperature of about 350 °C at atmospheric pressure. The residue of the first step is subjected to a higher temperature, around 350–425 °C, under a controlled pressure ranging from 1 kPa to 10 kPa [26]. The residue of the second process is called straight-run bitumen [27]. Moreover, if the residue of this second process is subjected to another process of thermal distillation at temperatures between 455 °C and 510 °C, visbreaker bitumen is produced [28]. Vis-breaking allows refineries to reduce the amount of the residue produced, as it allows the further recovery of lighter products such as diesel and gas. This penalizes the quality of the bitumen that is obtained, which is more rigid, brittle, and susceptible to aging [29,30].

### 2.1. Basic Characterization

The bitumen composition is strictly related to the characteristics of the starting crude oil, particularly on its age and the depth from which it is extracted. The study of bitumen chemical composition is very tricky because it contains many chemical elements. Although bitumen is mainly composed of hydrogen (8–12% by weight) and carbon (80–88% by weight), which together give a hydrocarbon content of about 90%, heteroatoms as nitrogen (0–2% by weight), oxygen (0–2% by weight), and sulfur (0–9% by weight) are also present. Moreover, there can also be traces of heavy metals such as nickel and vanadium, in the order of hundreds of parts per million [31].

The hydrocarbons (C+H) can be classified, according to the type of bond between the carbon atoms, into:

- saturated: only simple bonds are present between the carbon atoms;
- unsaturated: double or triple bonds are present.

The heteroatoms (N, S, O) can be found in the correspondence of the unsaturated bonds. When combined with carbon, these can cause an imbalance of the electrochemical forces that gives polarity to the molecule. Therefore, the heteroatoms, despite being present in small percentages, have the ability to make the unsaturated molecules more active, influencing the bitumen rheological properties [32].

### 2.2. Chemical-Structural Analysis—SARA Analysis

To interpret bitumen properties from its chemistry, it is necessary to consider it on different scales and not only at a global level. In fact, bitumen can be described as several central structures consisting of polyaromatic assemblies containing a various number of molten rings, saturated polycyclic structures, and combinations. Saturated hydrocarbon side chains, which are characterized by different dimensions and patterns, are linked to these central assemblies. Therefore, the number of possible isomers is almost unlimited. This is the reason why bitumen is characterized by millions of different molecules and none of these are present in such a high quantity to isolate and characterize them. Therefore, a chemical-structural analysis is more useful and appropriate to understand bitumen composition and mechanical behavior [33].

The chemical-structural analysis has progressed with fractionation techniques, through which it is possible to separate the bitumen molecules into chemical groups according to the dimensions or the soluble properties in various kinds of solvents (polar, apolar, or aromatic). Over the years, the fractionation techniques applied to bitumen have undergone a series of advances that have led to increasingly interesting results, clarifying more and more the structure of the material and therefore allowing a deeper understanding of the bitumen chemical composition.

In 1836, Boussingault separated two components of bitumen by distillation. He obtained two fractions named "petrolenes" (85% by weight) and "asphaltene" (15% by weight). Given the similar H/C ratio of the two fractions, he thought that asphaltene could derive from the oxidation of petrolene [34]. A few decades later, Richardson made his contribution, defining "asphaltenes" as the part of the bitumen insoluble in naphtha but soluble in carbon tetrachloride ($CCl_4$). Moreover, he introduced the "carbeni" and the "carboids", which are, respectively, soluble and insoluble in $CS_2$ [35].

Kayser, in 1897, used three solvents, chloroform, ether, and alcohol, to obtain three bitumen fractions [36]. Hoiberg achieved greater success in 1939 when he achieved the separation of the maltenes in resins (precipitate) and oils (soluble part) [37]. Corbett proposed a method to further split the maltenes and obtain three categories: saturated, aromatics, and resins [38]. Therefore, he managed to separate the bitumen into the four fractions that are still considered today in the chromatographic analysis: Saturated, Aromatic, Resins, and Asphaltenes (SARA); hence the SARA terminology is obtained by joining the initials of each fraction. Figure 2 summarizes the processes for SARA fraction separation.

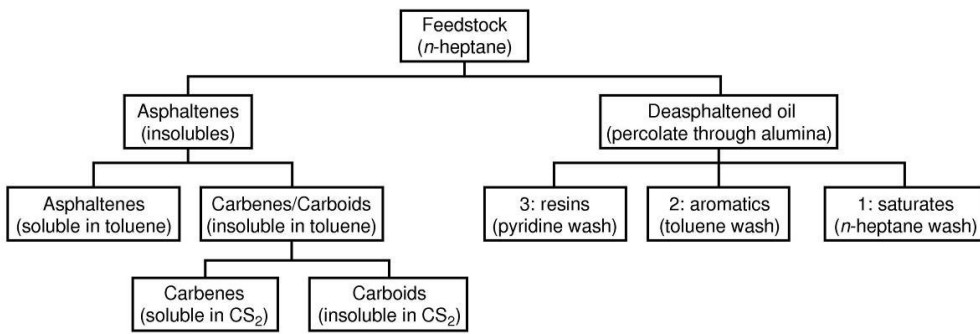

**Figure 2.** Bitumen composition (reprinted from Reference [39] with the permission of Elsevier).

Nowadays, the process defined by ASTM D-4124 [40] is divided into several steps. The first consists of the separation of asphaltenes with precipitation of n-heptane. Afterward, the maltenes in a solution of n-heptane are poured into a chromatographic column to separate the saturates; then, the aromatics are separated using 100% toluene and a 50/50 blend of toluene and methanol; finally, the resins are obtained using trichloroethylene.

Several methods are available to obtain the SARA fractions [41,42], but despite the proportions depending on the origins of the raw material, the use of different techniques gives slightly different results. Therefore, to compare different varieties of bitumen according to the SARA fractions, the use of the same fractionation method is always a good practice. A deeper description of the four distinct fractions is provided below.

Saturates represent about 5–15% by weight of bitumen and consist of an almost transparent liquid. They are mainly composed of aliphatic (branched, linear, and cyclic hydrocarbons) and have no polarity (rare aromatic rings or polar atoms). They function as a jellying agent for bitumen components, as they favor asphaltene flocculation and therefore the bitumen solid-elastic phase.

Aromatics represent about 30–45% by weight of bitumen and consist of a yellow-red oily liquid. They contain one or more aromatic rings and act as solvents for the peptized asphaltenes.

Asphaltenes represent about 5–20% by weight of bitumen and consist of a dark powder [43]. They have many aromatic rings and polar compounds. They are the main component associated with bitumen stiffness and viscosity [44].

Resins represent about 30–45% by weight of bitumen and consist of a black solid. They are similar to asphaltenes in terms of composition but they have a higher polarity [45]. They are the component that is associated with bitumen stability, as they behave as flocculent agents for the asphaltenes.

The bitumen behavior is determined by the relative amount of the components, but especially by the compatibility and the interactions among these homogeneous fractions.

### 2.3. Colloidal System

Several models have been proposed to understand the rheological properties of bitumen through its chemical composition. Although Rosinger had thought about a colloidal structure for bitumen in 1914 [46], today this intuition is attributed to Nellesteyn, who described the bituminous colloidal system in 1924 [47]. They described bitumen as the dispersion of asphaltene micelles (solid) in an oily phase (fluid) thanks to the presence of peptizing agents. The first description of the colloidal system is a structure determined by asphaltenes micelles immersed in a maltene solution. In particular, the asphaltenes are covered by the maltenic polar part (resins), which acts as a peptizing agent for the asphaltenes themselves, and everything is immersed in the so-called oils, flocculating agents for asphaltenes.

In the following years, Pfeiffer and Saal [48] introduced the difference between two colloidal systems (sol and gel), which represent the two colloidal limit systems for all bitumen.

In particular, if resins keep asphaltenes highly peptized (or dispersed) in the oily phase so that micelles are not interacting, the associated bitumen is characterized by a sol model. This model results in a very viscous (not elastic) behavior at low temperatures and a Newtonian liquid behavior at high temperatures. If resins are not very effective in peptizing asphaltenes, which become fully interconnected, a gel model is obtained. This model results in non-Newtonian fluid (viscoelastic) behavior at high temperatures and elastic solid behavior at low temperatures.

Most of the bitumen shows intermediate characteristics between these two structures, which represent the limit cases. The coexistence of the sol-type micelles and the gel structure as a function of temperature and aggregation state of micelles (ratio among asphaltenes, resins, aromatics, and saturates) is defined as a gel-sol model (Figure 3).

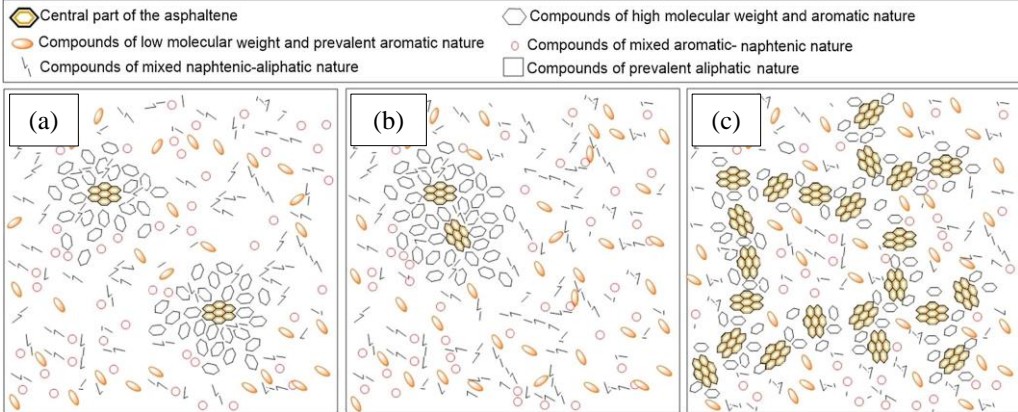

**Figure 3.** Schematic representation of the colloidal model of bitumen: (**a**) sol, (**b**) flocculated asphaltene micelles, (**c**) gel (reprinted from Reference [49] with the permission of Elsevier).

In 1971, Gaestel et al. [50] introduced the concept of the Colloidal Index (CI) or Instability Index, whose empirical expression is reported below.

$$CI = \frac{Asphaltenes + Saturates}{Aromatics + Resins}$$

Generally, this index has a value between 0.5 and 2.7 for the most used bitumen. The bitumen shows a clear gel behavior if the colloidal index is greater than 1.2, while the behavior is closer to a sol model if the colloidal index is lower than 0.7.

### 2.4. Test Methodologies to Investigate Bitumen Chemistry

Nowadays, a wide variety of methodologies are available to analyze the chemical properties of bitumen. Each technique has some issues, because the results are different as a function of the nature of the binder and the process conditions under which it is analyzed. Therefore, to deeply investigate the chemical properties of bitumen, it is a good practice to combine multiple chemical tests, together with a rheological and traditional characterization.

The most frequently used techniques for the chemical analysis of bitumen are summarized in Table 2 and described hereafter.

**Table 2.** Most frequently used techniques for the chemical analysis of a bitumen.

| Technique | Type of Analysis | Parameters Used |
|---|---|---|
| AFM: Atomic Force Microscope | Microscopic | Microstructure and micro-mechanical properties of bitumen |
| FTIR: Fourier Transform Infrared spectroscopy | Chemical | Quantity of carbonyl and sulphoxide groups |
| TLC-FID: Thin Film Chromatography with Flame Ionization Detection | Chemical | Saturated, aromatic, asphaltene and resin content |
| HP-GPC: Gel Permeation High-Pressure Chromatography | Chemical | Number of chemical groups and molecular weights |

### 2.4.1. Atomic Force Microscope (AFM)

The AFM test is a non-destructive analysis that allows representing of the surface morphology of a bitumen sample, as well as information regarding stiffness, cohesion, and molecular interactions at a microscopic level. The fundamental principle on which this test is based is very easy to understand. The device is equipped with a flexible cantilever, which is linked to a piezoelectric component and has a tip at the extremity. During the test, the tip slides on the bitumen surface while its position is measured through a laser system and the resistance to the tip movement, which depends on the distance between atoms, is registered. By coupling this information, a detailed scansion of the bitumen sample surface at a microscopic (atomic) scale is collected [51].

The AFM technique allows the identification of three major phases of different rheology and composition (Figure 4):

1. Catana phase or bee phase, which are a sort of hills in the undulated pattern of the AFM image;
2. Peri phase (from Greek peri = around), surrounding the bees and characterized by a certain roughness;
3. Para phase (from Greek para = neighbor), next to the peri phase and typically flat [52–54].

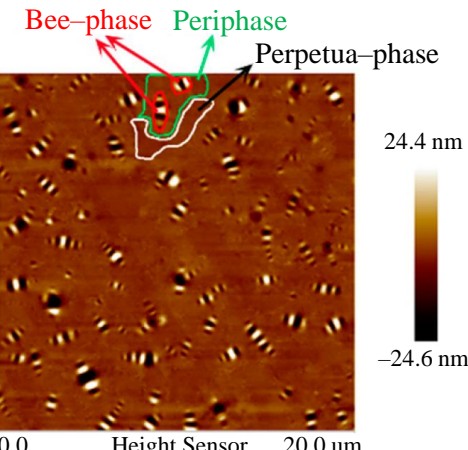

**Figure 4.** Three major phases of the superficial morphological representation of bitumen (reprinted from Reference [55]).

2.4.2. Fourier Transform Infrared (FTIR) Spectroscopy

FTIR spectroscopy is widespread technology for the identification and analysis of organic compounds. In particular, FTIR is a method for determining the molecular structure of a material by measuring the atom oscillations (rotations and vibrations). During the test, infrared radiations hit the sample, whose atomic functional groups absorb part of these radiations. In particular, the specific wavenumber of the absorbed radiation is a function of the vibration mode of the functional group. Through the application of the Fourier transform, the absorbance spectrum of the sample is obtained.

Each functional group has different vibration modes, whose number is related to the number of atoms and type of bond. For instance, the symmetrical molecules that include two atoms (e.g., diatomic nitrogen $N_2$) have no absorption in the IR spectrum, while asymmetrical diatomic molecules (e.g., carbon monoxide CO) do. For more complex functional groups, for example methylene (-$CH_2$), the vibration modes (Figure 5) include six types of oscillations [56].

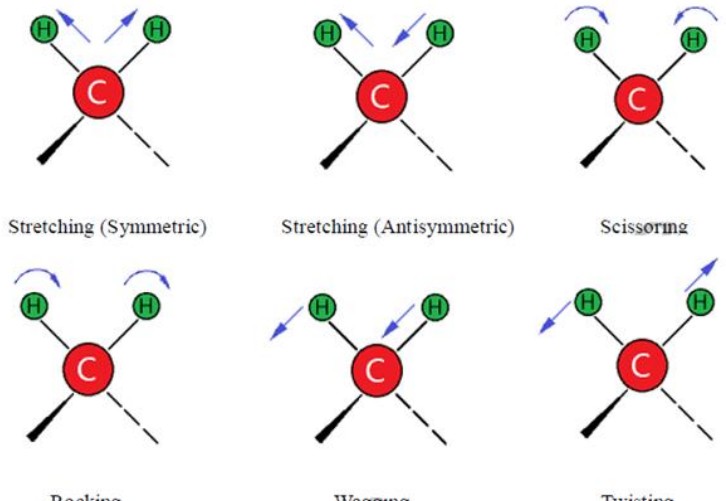

**Figure 5.** Vibration modes of the methylene group (-CH2) (reprinted from Reference [57] with the permission of Elsevier).

The bitumen characterization through FTIR spectroscopy provides different information. The analysis of the peak position in the spectrum allows the identification of the functional groups. Moreover, the height of the peaks allows the quantification of the functional group concentration [57].

### 2.4.3. TLC-FID: Thin Film Chromatography with Flame Ionization Detection

TLC–FID is used to quantify the content of each SARA fraction in a bitumen. The bitumen is initially blended with cyclohexane solvent and a small amount of solution is put on a quartz rod, known as a chromarod. Then, the procedure is repeated using three more solvents (n-hexane, toluene, dichloromethane). The separated fractions (i.e., asphaltenes, resins, aromatics, and saturates) are located in the chromarod series at, respectively, 0 cm, 2.5 cm, 5 cm, and 10 cm, as a function of the decreasing polarity. Finally, the chromarod is analyzed by Flame Ionization Detection (FID), which step-by-step ionizes the different zones corresponding to the four SARA fractions, and allows the estimation of their percentages in the bitumen [58,59].

### 2.4.4. HP-GPC: Gel Permeation High-Pressure Chromatography

HC-GPC is a test method that allows quantifying of the molecular size distribution of bitumen. The binder sample is dissolved in tetrahydrofuran (THF), and the solution is put in a column for chromatographic analysis. Gel (stationary phase) is used as a stabilizing agent that slows down the permeation of light components, while the high pressure allows increasing the test speed and the separation efficiency.

Typically, thirteen slices of the chromatographic pattern are assumed in order to discriminate between large (slices 1–5), medium (slices 6–9), and small (slices 10–13) molecular sizes [60].

The molecular weight can be represented in terms of average molecular weight by molecule weight ($M_w$) and average molecular weight by molecule number ($M_n$). In addition to $M_w$ and $M_n$, their ratio ($M_w/M_n$), defined as "molecular weight dispersion index", is also often considered in the study of bitumen molecular size distribution [51]

## 3. Bitumen Aging

Bitumen aging is defined as the series of chemical transformations that the material undergoes and that results in the variation of its physical characteristics [61]. In general, two different aging processes, namely short-term aging and long-term aging, are identified.

Short-term aging is the phenomenon that bitumen suffers during HMA manufacturing (mixing, hauling, paving, and compacting) because of the high processing temperatures (>150 °C). Long-term aging is the phenomenon that affects bitumen during the entire service life of the mix, which is subjected to traffic and environmental stresses. The severity of this process is mainly related to the bitumen's exposure to air, which depends on the mix air voids and the position of the HMA layer within the pavement structure [39].

Despite some studies [62,63] that have identified several variables that affect bitumen aging, the most widely recognized mechanisms include:

- Physical and steric hardening (reversible mechanisms);
- Loss of low-weight components (volatiles) by evaporation;
- Oxidation, with the consequent changes at the molecular level that cause a change in the SARA fractions.

The oxidation and evaporation of volatiles, which are irreversible processes, are accelerated during the HMA production and paving when the bitumen is hot [64]. When the mix reaches the air temperature, the evaporation of volatiles becomes much less influential, while oxidation continues in the long-term aging. Furthermore, the greater importance of oxidation compared to that of physical hardening is given by the different nature of these processes: oxidation is irreversible, while physical hardening can be recovered. For these reasons, oxidation is considered the main process in the aging of bitumen.

These mechanisms are described in detail in the following sections.

### 3.1. Physical and Steric Hardening

Physical hardening deals with the changes in the bitumen viscoelastic properties due to the material cooling below the glass transition region. However, the process does not entail any change in the chemical structure and is reversed when the bitumen is re-

heated to air temperatures [65]. Assuming that the material volume consists of the volume of the oscillating molecules and the free volume between the molecules [66,67], when the bitumen temperature decreases, both molecular mobility and free volume reduces, maintaining the same proportion between occupied and free volume. When the glass transition temperature is reached, the free volume decrease becomes slower than the decrease of molecule oscillation, entailing a kind of "over-hardening" for the bitumen [68].

Physical hardening should not be confused with steric hardening. Steric hardening is a chemical process in which the bitumen molecules rearrange and form wax compounds in the maltenes, due to the presence of linear alkanes in the asphaltenes [69]. The process happens at intermediate temperatures but takes three times longer than physical hardening [70]. Even steric hardening is reversible. While the former type of hardening occurs within 1–2 days at temperatures below the glass transition temperature of bitumens (−35/−15 °C), the latter is manifested at room temperature, and requires days or even weeks. The steric hardening is related to the inner reorganization of the binder molecules; it is associated with the formation of ordered structures by waxes in the maltenes phase that is influenced by the linear alkanes present in the asphaltenes fraction. It is a reversible process because it can be removed by heating or mechanical work [71].

### 3.2. Evaporation of the Volatile Components

The evaporation of saturated and aromatic components has been also reported as an aging mechanism of bitumen. In particular, this phenomenon is mainly related to short-term aging, as it depends on the temperature to which the bitumen is subjected during the mixing and installation phases [72]. It has been quantified that the loss of volatiles can be double for a temperature increase of 10 °C during HMA manufacturing at the plant [73]. The volatile evaporation causes the unbalancing of the SARA fractions, determining the predominance of resins and asphaltenes over saturates and aromatics. Consequently, the bitumen that results is harder, stiffer, more viscous, and more fragile. The evaporation of volatile compounds is an irreversible mechanism that significantly affects bitumen aging, even if to a lower extent than the oxidation process [74].

### 3.3. Oxidation

Thurston and Knowles, in 1941, demonstrated how bitumen components, in particular asphaltenes and resins, absorb oxygen [63]. It is widely accepted today to consider the oxidative process of bitumen as the most important mechanism that happens during aging.

Bitumen oxidation is an irreversible process that deals with the "capture" of oxygen atoms by the bitumen components (particularly the asphaltenes), which undergo an alteration of their chemical characteristics. Moreover, this aging process could be photo-catalyzed in the case of the bitumen in pavement surface layers, in particular for polymer-modified binders [23]. As oxidation is due and depends on the access to oxygen in the mixture, the voids content, the HMA layer depth, the bitumen content, and the presence of cracking are factors that can influence the quantity of bitumen exposed, and therefore the quantity of potentially aged bitumen.

Within bitumen morphology, oxidation includes dehydrogenation, the reaction of the alkyl sulfides into sulphoxides, and the reaction of the benzyl carbons into ketones, which in turn form carboxylic acids with dicarboxylic anhydride. These reactions can be quantitatively determined by functional group analysis through FTIR spectroscopy. A typical infrared spectrogram is shown in Figure 6. The absorbance bands around 1690 cm$^{-1}$ are due to the increase in C=O bonds (carbonyl groups)—for example, ketones, carboxylic acids, and anhydrides—while those around 1030 cm$^{-1}$ are due to the increase in S=O bonds (sulphoxide groups). Consequently, the peak areas of the two wavenumbers can be considered as concentration measurements of carbonyl compounds and sulphoxides, respectively [75].

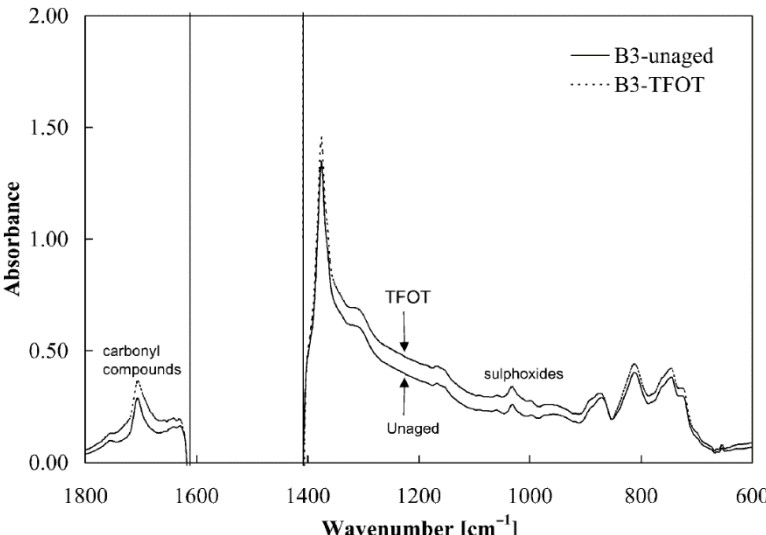

**Figure 6.** Effect of aging on bitumen FTIR spectrogram (reprinted from Reference [75] with the permission of Elsevier).

It is important to highlight that the carbonyls, ketones, and sulfoxides generated through oxidation are characterized by a marked polarity. Therefore, they associate with the polar groups in the bitumen forming agglomerates with a high molecular weight [76]. These large and "heavy" clusters, which typically involve the asphaltene fraction, determine the reduction of the molecular mobility within the bitumen colloidal system, resulting in increased viscosity, stiffness, and brittleness [77].

Temperature is a key factor in the oxidation phenomenon. In particular, the degree of oxidation is doubled every time the temperature increases by 10 °C (after 100 °C) [73]. The influence of temperature was also observed in the laboratory by Lu et al. [78]. They noted that it takes 4–8 longer times to obtain the same aging when the pressure aging vessel (PAV) temperature is decreased from 100 to 75 °C.

As previously said, the oxidation reaction can be promoted and accelerated when the bitumen functional groups are excited by UV radiations. This issue has been neglected for many years as it only involves the upper pavement layer, because of the low ability of the radiations to penetrate in depth. However, the amplifying effect on the aging of the bituminous binder due to ultraviolet radiation should be considered, particularly in the most exposed surfaces of geographic regions characterized by high levels of solar radiation and humidity [79]. Many authors associated the increase in bitumen viscosity, as a consequence of oxidation, with the number of radiations that invested the material [80–82]. In particular, Zeng et al. [81] observed the detrimental effects of ultraviolet radiation in association with high temperatures in promoting oxidation. Afanasieva et al. [82] specified that bitumen is highly oxidized when subjected to radiations with a wavelength coinciding with the UVB range (280–315 nm).

### 3.4. Laboratory Aging Methods

To date, the most frequently used methodologies to age bitumens are the thin film oven test (TFOT), rolling thin film oven test (RTFOT), pressure aging vessel (PAV), and ultraviolet test (UV). Most of them are often characterized by increases in temperature, oxygen pressure, or a combination of these two in order to generate the aging conditions, which are as close as possible to the conditions in which real bitumen can be found. While the first two methodologies are mainly adopted to reproduce short-term aging, which occurs during storage, mixing, hauling, and laying of an HMA, the last ones can simulate long-term aging that occurs during the service life of the pavement.

### 3.5. Laboratory Aging Assessment Methods: General Results

As regarding the different SARA fractions, aging can generally be summarized as follows:

- the saturates remain almost unchanged;
- the aromatics decrease;
- the resins see a small increase;
- the asphaltenes increase.

In particular, the general effect of oxidation is the shift of each SARA fraction towards the next component in the polarity scale. As explained before, the four bitumen fractions are ranked, as a function of the increasing polarity, as follows: saturates, aromatics, resins, and asphaltenes. Actually, the saturates only show little changes between the unaged and long-term-aged binder, so they can be considered an unreactive fraction [27]. The next fractions, aromatics and resins, oxidize and respectively shift into resins and asphaltenes. Since aromatics evolve into resins, but there is poor supply from the saturates (which are mainly inert), a global decrease of the aromatic content with aging is observed. Differently, the resin content only experiences a slight increase or decrease since there is the contemporary uptake of the oxidized resins into asphaltenes and of the oxidized aromatics into resins [83].

The relationship between resins and asphaltenes plays a crucial role in aging: the asphaltene fraction is the component that grows the most; at the same time, the resin content increases to a lesser extent, facilitating the mutual contacts between asphaltenes. When the ratio between aromatics and resins is not high enough to allow the peptization of the asphaltene micelles, or when the solvation capacity of the system is insufficient, the micelles tend to bond to each other [84]. This determines the formation of larger irregular structures in which voids are present (filled by the external liquid of the component micelles). Thus, in terms of colloidal models, aged bitumen tends to assume a gel structure, causing a stiffer and more brittle behavior [85]. The decrease of the maltenes affects the stability of the colloidal system and determines the flocculation of the asphaltenes. Moreover, the higher amount of asphaltenes increase the propensity of the asphaltenes themselves to micellization and agglomeration [86].

Li et al. [51] quantified the variation of SARA fractions during aging. In this study, five different bitumens were separated into the SARA fractions before and after aging. Figure 7 highlights how, during aging, the asphaltenes increase while the contents of aromatics and resins decrease. Moreover, the amount of saturates tends to remain stable. This phenomenon was investigated in terms of the colloidal index (CI), which proved to grow with aging, particularly in the long-term step.

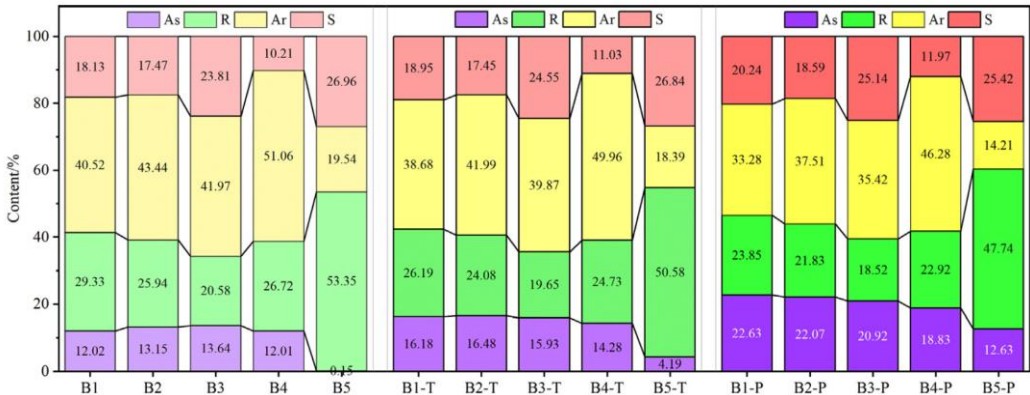

**Figure 7.** Proportion of bitumen SARA fractions with aging (reprinted from Reference [51] with the permission of Elsevier).

Mirwald et al. [83] tried to evaluate (using FTIR), not only the changes in quantities but also the chemical modifications within individual SARA fractions during aging. In particular, an unaged bitumen and three long-term-aged bitumens were separated into

their SARA fractions, which were subsequently analyzed through FTIR spectroscopy. The absorbance spectra showed that:

- The saturates spectrum remains unchanged, confirming that this fraction is little affected by aging;
- The aromatic spectrum shows some changes (an increase of the peaks in the carbonyl band, main aromatic band and across the entire fingerprint area), but no significant increase in sulfoxides;
- The interpretation of the resin spectrum is complex, because of the position of this fraction in the polarity gradient. In general, aging results in uptake from 2-quinolones and carbonyls into ketones and in the growth of the sulfoxides content;
- The asphaltenes spectrum shows significant variations in the fingerprint area. In particular, the aging determines an increase in sulfoxides ($1030 \text{ cm}^{-1}$), main aromatic ($1600 \text{ cm}^{-1}$), and carbonyl ($1700 \text{ cm}^{-1}$) peaks.

Thus, the increase of the carbonyls affects the aromatics and the resins, while the increase of the sulfoxides affects the resins and the asphaltenes.

Using GPC, Li et al. evaluated the molecular weight of the bitumen components [51]. Table 3 shows that the molecular weight increases when moving from saturates, to aromatics, to resins and asphaltenes. In particular, the saturates have the shortest molecular chains, denoting an approximate structure. Differently, the asphaltenes have the longest chains and the most complex structure. In addition, they include a higher number of polar groups that have a low sensitivity to temperature changes. Therefore, when the asphaltene content increases, the bitumen tends to maintain its mechanical properties (stiffness, viscosity), even when increasing temperature [51].

**Table 3.** Molecular weight of the bitumen SARA fractions (reprinted from Reference [51] with the permission of Elsevier).

| Component | $M_n$ | $M_w$ | $M_w/M_n$ | Features |
|---|---|---|---|---|
| Saturated (S) | 506 | 673 | 1.329 | Low molecular region |
| Aromatics (Ar) | 648 | 1220 | 1.882 | Transition region |
| Resins (R) | 907 | 2761 | 3.045 | Transition region |
| Asphaltenes (As) | 1898 | 14,660 | 7.725 | High molecular region |

FTIR spectroscopy has been used for many years to analyze the bituminous binders [87], particularly to investigate the polymer modification mechanisms and the effects of aging. Table 4 summarizes the main changes noticed in the bitumen FTIR spectrum as a consequence of aging.

**Table 4.** Change of key FTIR bands during aging.

| Chemical Group | Bond | Approximate Wavenumber ($cm^{-1}$) | Change with Aging | References |
|---|---|---|---|---|
| Sulphoxide | S=O | 1030 | Increase | |
| Carbonyl | C=O | 1690 | Increase | [57,88–92] |
| Aliphatics (plan deformation) | $CH_2$, $CH_3$ | 1460, 1375 | Small decrease | |
| Aromatics | C=C | 1600 | Small increase | [88,90,93] |
| Aliphatics (asymmetric or symmetric stretching | $CH_2$, $CH_3$ | 2923, 2853 | Small decrease | [93,94] |
| Polarity | O–H | 3450 | Increase | [93,94] |

The more significant changes in the FTIR spectrum associated with bitumen oxidation are the rise carbonyl C=O ($1690 \text{ cm}^{-1}$) and sulfoxide S=O (at $1030 \text{ cm}^{-1}$) bands. This has

been observed in both laboratory [90,95] and site [88,89] aged bitumen. In particular, the work by Mouillet et al. [90] specified that increases in the S=O and C=O bands are mainly related to short-term aging and long-term aging, respectively. Even the aromatic C=C band (1600 cm$^{-1}$) shows a slight increase that is associated with the increase of the resins and especially the asphaltenes, which include condensed aromatic rings. The polarity band (around 3450 cm$^{-1}$) is highly marked in the spectra of resin and asphaltene components, but this has not been exactly associated with the bitumen aging process [93]. Finally, the aliphatic CH$_2$ and CH$_3$ bands showed a slight decrease with increased aging [94].

In order to quantify the effects of bitumen oxidation, two indices have been introduced:

- Carbonyl index: $I_{C=O} = \frac{A_{1690}}{A_{ref}}$
- Sulphoxide index: $I_{S=O} = \frac{A_{1030}}{A_{ref}}$

where A$_{1690}$ is the area of the C=O peak centered at 1690 cm$^{-1}$, A$_{1030}$ is the area of the S=O peak centered at 1030 cm$^{-1}$, and A$_{ref}$ is the area of the reference ethylene and methyl peaks, centered at 1460 and 1375 cm$^{-1}$, respectively [96].

With the aim to estimate the effects of bitumen aging on its properties, the variation of the indices in unaged, short-term-, and long-term-aged bitumen can be calculated. When increasing the aging, the heights and the areas of the peaks in correspondence of the wave numbers of 1690 cm$^{-1}$ and 1030 cm$^{-1}$ (respectively for the carbonyl C=O and sulphoxide S=O bands) increase, so the two indices also increase [97–100]. Moreover, a recent study [101] proposed the Chemical Aging Index (CAI), calculated as $I_{C=O}$ plus $I_{S=O}$, in order to better understand the variation of both indices during aging.

Regarding the AFM test, changes of the particular bee-shaped structure can be used to investigate the aging process. Nowadays, there is still some discussion about the nature and conformation of these structures. There is a certain agreement that the bee structures are associated with wax crystallization [102]. For this reason, it was hypothesized that the "bees" are correlated to the asphaltenes [103,104]. A study by dos Santos et al. [105] demonstrated that the valleys, due to their lower thickness and roughness, are less strong than the hills. Other studies proposed that the formation of the bee structure is associated not only with asphaltenes but also with resins and aromatics, and so it is strictly correlated to the bitumen performance [106,107]. One more point of discussion is related to the presence of these phases in the whole bitumen volume or only on the surface [108].

Li et al. [51] tried to identify the changes in the bee structures during short- and long-term aging in different bitumens. Before aging, these show the distinct elliptic bee structures, which are short and thick. After aging, different behaviors were observed for the various bitumens. In general, the bee structures became larger and irregular when increasing the aging level. In addition, peculiar phenomena happened in the AFM diagrams, such as the increase of the bee stripes number or the formation of sunk regions, columnar peaks, blocky structures, or cracks.

A study by Lu et al. [109] investigated wax-including bitumens in comparison with a non-waxy bitumen using AFM. They noted that aging led to a decrease in the number and an increase in the dimensions of the bee structures in the bitumen with wax, probably because of the highly increased stiffness of the aged binder and/or the reduced compatibility between the saturated crystalline fraction and the more polar bitumen matrix. On the other hand, for a non-waxy bitumen, no structure was observed either on the unaged or PAV-aged state. Moreover, after 60 h of PAV aging, the content of asphaltenes increased from 22% to almost 25%. This demonstrated that asphaltenes are not the fraction responsible for the structures, unless they contain n-heptane insoluble crystallizable materials.

Zhang et al. [110] also studied the aging effect on the bee structures. They confirmed that aging entails an increase in the bee structure number, dimension, and roughness (Figure 8), which is associated with a contemporary decrease of the penetration and increase of the ring and ball softening point and viscosity.

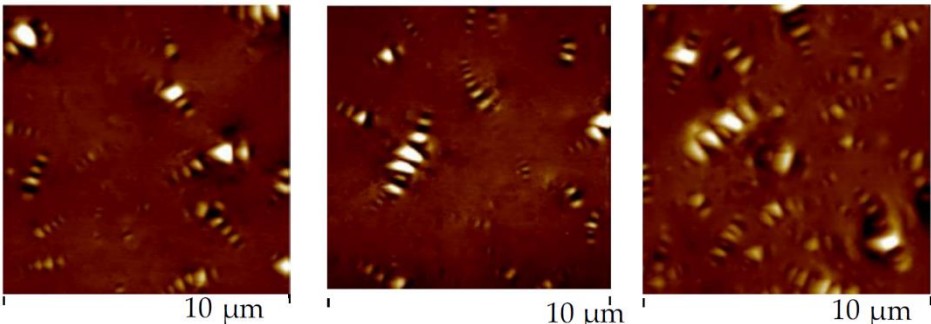

**Figure 8.** AFM images of unaged, short-term-aged, and long-term-aged bitumen (reprinted from Reference [110]).

## 4. Bitumen Rejuvenation

Including RAP in HMA requires a careful study of the mixture, because its mechanical properties are strongly conditioned by the presence of the aged bitumen contained in RAP. In particular, improper use of RAP can lead to premature cracking linked to the excessive stiffness of the bitumen (thermal and fatigue cracking) [111–114]. In order to achieve the proper mechanical properties for the HMA, additives are included in the mixture. In the following paragraph, the concept of rejuvenator is provided, clarifying which are the main distinctions within this category and explaining the benefits associated with their use.

A rejuvenator is an agent that allows renovating the properties that the bitumen loses with aging [115]. Thus, rejuvenators should reduce aged bitumen viscosity and stiffness and improve ductility [24]. However, the many products that can be used for this aim may act at a different level, for example, in the colloidal system or the chemical morphology. The literature does not seem very clear about this; the term "rejuvenator" is often used for any additive, without specifying what the effective mechanisms of action in bitumen are. According to different authors [24,116,117], rejuvenators can be classified, based on the effect, as:

- Softening agents (also called fluidifying agents or rheological rejuvenators), which include:
  - Incompatible softeners, which mainly have a viscosity lowering effect;
  - Soluble softeners, which restore the balance in the SARA composition by re-enriching the maltene fraction;
- Real rejuvenators or compatibilizers, which help to renovate the physical and chemical characteristics of the bitumen through the disruption of the intermolecular associations between the asphaltenes.

The softening agents are usually based on extracts of lubricating oils. Slurry oil, flux oil, and lube stock belong to this category. They include a proper content of maltenic, naphthenic, or aromatic components aiming to rebalance the SARA fractions of the aged binder, characterized by a lower concentration of maltenes. The softening agent allows an increase in bitumen ductility and a decrease in viscosity and brittleness by merely supplying oily components to maltene fraction, but does not achieve any change in the complex structure. The real rejuvenators are additives that allow restoring both the physical and chemical properties of bitumen. In particular, they are able to restore the agglomeration condition of asphaltenes to their original state by promoting their re-dispersion. Regarding their composition, real rejuvenators should have a high content of aromatics, which allows keeping the asphaltenes dispersed and maintains a low content of saturates, which have poor compatibility with the asphaltenes.

The efficacy of the rejuvenation process is strictly related to the adequate dispersion within the aged bitumen. Lee et al. first faced this aspect, claiming that a mechanical mixing could allow a uniform distribution [118]. A few years before, Carpenter and Wolosick divided the diffusion of a rejuvenator into an aged binder into four different steps. In the first phase, the rejuvenator forms a low-viscosity layer around the particles of

aggregates covered with aged bitumen. Then, the additive starts to penetrate the binder by softening it. In the third phase, the rejuvenator penetrates the aged bitumen, and the viscosity of the internal and external surfaces gradually decreases. Finally, in the last phase, over time, the rejuvenator manages to reach all the aged bitumen [119]. Noureldin and Wood [120] and Huang et al. [121] confirmed this theory some years later. Karlsson and Isacsson [122], found out that the diffusion of the rejuvenator into the aged bitumen is mainly influenced by the viscosity of the maltene phase. Therefore, it can be facilitated by raising the temperature or adding oil [123].

### 4.1. Rejuvenators for Hot Recycled Mix Asphalt (HRMA)

During the last decades, many studies have tried to estimate the efficacy of different rejuvenators. According to the literature, there are many different sources from which we can obtain rejuvenators. In general, according to their nature, rejuvenators can be classified as additives derived from oil and biological additives. For instance, Zaumanis et al. [124] and Dony et al. [125] stated that although the two categories may show differences, both additives can be used successfully to soften the aged bitumen and allow the fulfillment of the requirements in terms of penetration, softening point, and rheological characteristic.

Besides fuels and bitumens, rejuvenators able to improve the properties of an aged binder are often produced from the processing of crude oil. These products include aromatic extracts, paraffinic oils, naphthenic oils, and spent motor oils [126]. These petroleum products have been the main agents used for many years to improve the characteristics of aged bitumen. In recent decades, research has moved towards finding materials with rejuvenating effects for aged bitumen that have lower costs and are environmentally sustainable. These are called bio-rejuvenators, and they are products such as tall oil, rapeseed oil, soybean oil, sunflower oil, corn oil, used cooking oil, castor oil residues, and organic oils. Several studies promote these rejuvenators, highlighting that they are able to restore the original properties of bitumen, while being eco-sustainable at the same time.

Among the numerous studies carried out during the last decades, the most relevant and suitable are reported hereafter, together with a brief description of the used rejuvenators and the main results.

Bocci et al. [127,128] and Grilli et al. [129] focused on the evaluation of the mechanical and performance characteristics of HMA made with a high amount of RAP and a bio-rejuvenator. They found out that, using appropriate dosages of this additive, acceptable performance, similar to that reached with virgin materials, can be achieved. Thus, the evaluation of the correct dosage of rejuvenators plays a fundamental role in obtaining a mixture that can easily meet the specifications.

Krol et al. [130] and Somé et al. [131] evaluated the effects of several vegetable oils (rapeseed oil, soybean oil, sunflower oil, flaxseed oil) on the mechanical characteristics of bitumen and mixes. Furthermore, they also developed chemical processes to make new additives. These studies aimed to improve the performance of bituminous binders using only very cheap raw materials.

Zargar et al. [132] studied the possibility of using waste cooking oil (WCO) as a rejuvenator in aged bitumen. They highlighted that by adding a specific dosage of this additive, the same values of penetration index, softening point, and viscosity of a virgin bitumen can be obtained. Moreover, increasing the amount of WCO, these properties can be further improved. Gökalp and Emre Uz [133] confirmed that using WCO leads to improvements in penetration index, viscosity, and fatigue resistance.

During the last years, many studies have compared the effect of WCO with respect to waste engine oils (WEO) in restoring the original properties of an aged bitumen. In particular, Joni et al. [134] stated that the effect of WCO is greater than that of WEO on the properties of an aged binder; therefore, to obtain the same results, the required amount of WEO to be added is greater than that of WCO. Li et al. [135] confirmed that both WCO and WEO have rejuvenating properties, but WCO has higher efficacy than WEO, which implies the use of higher dosages. Moreover, Al Mamun et al. [136] investigated different contents

of RAP and WCO/WEO rejuvenators. Differently from the previous studies, they noted that WEO allows a higher reduction of indirect tensile strength and stiffness with respect to WCO under the same dosage, but WCO allows a higher content of RAP to be recycled.

The differences between bio-additives and aromatic additives were also studied from a molecular point of view by FTIR. Noor et al. [92] showed that using biological additives involves an increase of particular functional groups (C=O) in bitumen. Moreover, Cavalli et al. [137] stated that using rejuvenators such as seed oil or tall oil can increase the carbonyl and sulphoxide indices since they contain functional groups C=O and S=O. Investigating the performance of these indices is a good way to qualitatively understand the effects of a rejuvenating agent on an aged binder. Zhang et al. [138], through the analysis of the carbonyl, sulphoxide, and aromatic indices, assessed the degree of aging and rejuvenation of some mixtures including bio-additives.

Zeng et al. [139] tested castor oil as a rejuvenating agent, and they found out that using this additive, it is possible to obtain an improvement of the rheological properties of the aged binder.

The studies carried out by Elkashef et al. [140] and Nayak and Sahoo [141] focused on the use of soybean oil and Pongamia oil. They concluded that both products are good rejuvenators, capable of restoring the rheological properties that the binder lost with aging.

Kehzen et al. [142] tried to evaluate tung oil (also called "China wood oil") as an additive, showing how its addition improves the elasticity of an aged binder. In addition, with suitable amounts of tung oil, good performances of the mixture at high temperatures are also ensured.

Several studies regard the effects of rejuvenators engineered and "built" in the laboratory. In particular, Zhang et al. [143] tested a rejuvenating agent made of rubber oil, plasticizers, and surfactants, and demonstrated that the blend of these ingredients allows the binder to recover the malleability and ductility lost during the aging processes. Rzek et al. [144] obtained a rejuvenator by modifying pyrolytic condensate of scrap tires with tire crumb. The results confirmed that the addition of this product improved the properties of the binder. Moreover, mechanical and rheological tests showed that the amount of RAP can reach up to 60% when this engineered rejuvenator is added to the mixture.

Zaumanis et al. [124,145,146] tested the rejuvenating effect of many products of both biological and hydrocarbon nature, such as cotton seed oil, vegetable oil, used cooking oil, used motor oil, aromatic extract, distilled tall oil, distilled tall oil, exhausted motor oil, naphthenic oil, and aromatic extract.

Radenberg et al. [147] carried out a study testing 21 different types of rejuvenators commonly used in the road sector; the products were distinguished between "rheologically effective" and "chemically effective". While the former led to an increase in the maltenic phase, the latter reversed the effects of the oxidation process in the agglomerated compounds.

Table 5 presents recent publications concerning the use of additives to rejuvenate the aged bitumen in RAP. For a correct understanding, a brief description of the nature of the additives tested is also included.

**Table 5.** Most used additives, biological (green) and derived from oil (gold), used to rejuvenate aged bitumen.

| Additive | Dosage * | Description | References |
|---|---|---|---|
| Tall oil | 4–20% | Tall oil is an organic product deriving from the kraft process, a procedure for converting wood into wood pulp, the main component of the paper. It contains fatty acids, acid resins, and surfactants. | [124,127,145,146,148,149] |
| Exhausted vegetable cooking oil (mix of the main oils used for frying) | 1–20% | The chemical composition of these additives mainly contains fatty acids and methyl esters, with both oleophilic and hydrophilic properties. | [92,132,133,135,136,145,146, 150–155] |
| Sunflower oil | 5–9% | It is the oil extracted from sunflower seeds. Contains triglycerides, with a high content of linoleic acid. It has a high content of polyunsaturated fatty acids. | [130,131,137] |
| Linseed oil | 6–9% | It is the oil obtained by squeezing previously dried or toasted flax seeds. It is mainly composed of triglycerides. It is one of the vegetable oils with the highest concentrations of acidolinolenic acid. | [130,131] |
| Soybean oil | 6–9% | It is obtained by extraction from soybeans through a special process called "crushing" with the use of chemical solvents. It too is mainly composed of triglycerides. | [130,131,140] |
| Rapeseed oil | 1.5–9% | It is a vegetable oil produced from rapeseed seeds. It occurs naturally in many varieties. The resulting oil, therefore, depends on the characteristics of the rapeseed from which it is extracted. The chemical composition includes fatty acids and methyl esters. | [130,131,156] |
| Castor oil | 5–50% | It is a very valuable vegetable oil, which is extracted from the seeds of the castor plant. It is mainly composed of acylglycerides, and the main fatty acid present is ricinoleic acid. | [139,141] |
| Pongamia oil | 5–15% | It is a fixed oil derived from the seeds of the Millettia pinnata tree. Typically, Pongamia oil is made up of glycerides, especially triglycerides. It is considered a fluxing agent rather than a rejuvenator. | [141] |
| Tung oil | 2–8% | Also called China wood oil, it is the oil extracted from Aleurites fordii seeds. It is mainly composed of triglycerides and is considered a drying oil, with extremely short polymerization times. | [142] |
| Cashew oil | 5% | It is an oil that derives from natural resins that fill the interstitial spaces of the honeycomb structure of the cashew shell. The resin is made up of 80–85% of anacardial acids (o-pentadeca dienylsalicylic acid) and the remaining fraction is cardol and methylcardol. | [137] |
| Corn oil | 1.5–9% | It is an oil extracted from the germs of the seeds of Zea mays, a graminaceous plant native to North America. It has a composition similar to sunflower oil, very rich in linoleic acid. It is mainly composed of triglycerides | [157] |
| Cotton seed oil | 12% | It is the vegetable oil extracted from the seeds of cotton plants. It is mainly composed of triglycerides. | [124] |
| Oleic acid | 2.5–4.5% | It is an 18-carbon monounsaturated carboxylic acid of the omega-9 series. In the form of triglyceride, it is an important component of animal fats, and is the most abundant constituent of the majority of vegetable oils. | [158] |
| Organic oil from wood waste | 2–12.4% | A very wide range of types of timber can be used, such as Red Maple, Magnolia, Balsam, Poplar, Linden, Beech and Pine. | [159,160] |
| Vegetable waste fat | 12% | Material composed of waste grease produced by catering processes. | [145] |

**Table 5.** *Cont*.

| Additive | Dosage * | Description | References |
|---|---|---|---|
| Pig manure | 2–10% | It is the product of the fermentation of pig manure mixed with solid material used as bedding. | [159–161] |
| Algae additive | 10% | This is a bio-oil extracted from algae leaves or blooms through pyrolysis, and it is rich in phenolic compounds. | [161] |
| Waste engine oil | 1–20% | It is the waste lubricating oil used by engines. It is mainly produced from paraffinic oil. | [124,125,134–136,146,162–164] |
| Rubber powder from pyrolysis of used tires | 5–12% | Pyrolysis is a thermochemical decomposition process of organic materials, obtained by applying heat in the complete absence of an oxidizing agent. The pyrolytic product from tires pyrolysis contains high concentrations of polycyclic aromatic hydrocarbons. | [144,165] |
| Aromatic extract | 5–9% | Aromatic extracts are refined products from crude oil and constitute one of the most traditional classes of rejuvenators. Their chemical structure includes aromatic polar rings. | [124,146,166,167] |
| Naphthenic oil | 50–400% | Naphthenic oils are high-quality pure naphthenic mineral bases, obtained by hydrogen refining of selected crude oil. | [146] |
| "Soft" bitumen | 5% | Bitumen with a high penetration value and low stiffness. It is typically classified as a fluxing agent since it does not restore the physical and chemical properties of the aged binder. However, this binder can lead to a decrease in bitumen blend viscosity. | [168,169] |

* The dosages refer to the weight of the aged bitumen.

### 4.2. Rejuvenating Mechanisms

As previously explained, aging determines morphological changes in the bitumen: in particular, it causes more intense molecular interactions by introducing polar groups, which lead to an increase of the colloidal agglomerates into the bitumen volume. As the molecular structure strongly influences the rheological properties, any alteration of the balances and interactions of the polar and non-polar components in the bitumen can determine a modification of the thermal and mechanical properties in a mixture.

From a chemo-morphological point of view, rejuvenation is the inverse process of aging. True rejuvenation breaks the molecular aggregations and rebalances the SARA fractions, leading to an improvement in the rheological properties of the bitumen [170]. Remembering the classification made in the previous section, real rejuvenators are additives capable of both replenishing the lost volatile components and flaking the large agglomerates of asphaltenes into much smaller compounds. The studies by Pahlavan et al. [161,170,171], which explain in detail the effect of rejuvenation in the aged bitumen morphology, are summarized hereafter.

The approach adopted by Pahlavan et al. to investigate the rejuvenating process is based on molecular dynamics (MD) simulation and quantum mechanical studies through density functional theory (DFT). It starts from the analysis of the asphaltene monomer and the related oxidized dimer, which consists of a structure resulting from the interaction of two monomers of asphaltenes.

The electrons of the C=O groups located around the oxidized asphaltenes (red dots in Figure 9) change the distribution of $\pi$ electrons on these flat-shaped molecules. This determines an insufficiency of electrons at the $\pi$ bonds in the central part of the asphaltenes with a consequent decrease of the repulsive forces between the asphaltenes. This decrease leads to the formation of the dimer (two asphaltene monomers).

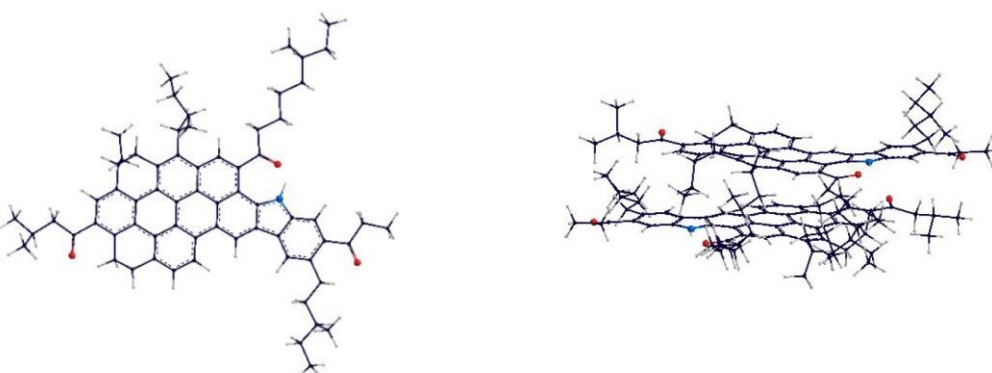

**Figure 9.** Asphaltene monomer (**left**) and dimer (**right**) molecular structures in the oxidized state (reprinted with permission from [161], Copyright 2019, American Chemical Society).

The forces that control the degree of asphaltenes stacking are concentrated in the aromatic motifs present in the asphaltenes. Figure 10 shows the most probable sites of interaction between a rejuvenator and an asphaltene dimer. The zones A, B, and C represent the areas where the rejuvenator likely finds a lower opposition by the hydrocarbon chains when it approaches the dimer. In these regions, the rejuvenation process happens in two steps that the author named the "lock and key" step and "intercalating" step. The first step ("lock and key") relates to zones A and B, where the adhesion between the asphaltenes in the dimer is stronger due to the polar groups. When the rejuvenator reaches the zones, it detaches the aromatic ring planes allowing access to more additive. In particular, the rejuvenator can intercalate in zone C, previously less accessible, and further deagglomerate the dimer by physically imposing a hindrance between the asphaltenes. The effectiveness of this second step depends on the chemistry of the rejuvenator. If the additive has polar properties, it creates interference on the Van der Waals forces that keep the polyaromatic cores stuck. However, the efficacy is higher if the additive has hydrocarbons with donor CH sites, which can interact with the $\pi$ electrons in the asphaltenes and thus disrupt the bound of the dimer.

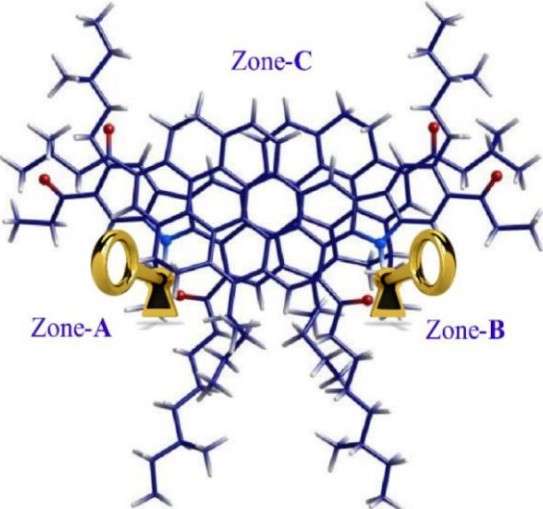

**Figure 10.** More likely interaction sites for biological rejuvenator with an asphaltene dimer (reprinted with permission from [161], Copyright 2019, American Chemical Society).

Of course, the rejuvenating effect of an additive is higher if it achieves both the steps, in particular, because the "intercalating" is noticeably facilitated when preceded by the "lock and key".

Figure 11 shows the side views of the dimer before and after rejuvenation. The simulation allowed observing an increase in the distance between the asphaltene sheets when the rejuvenator is used. In particular, in zones A and B (on the right in the side views), the increase of the gap between the polyaromatic cores was higher than in zone C (on the left in the side views).

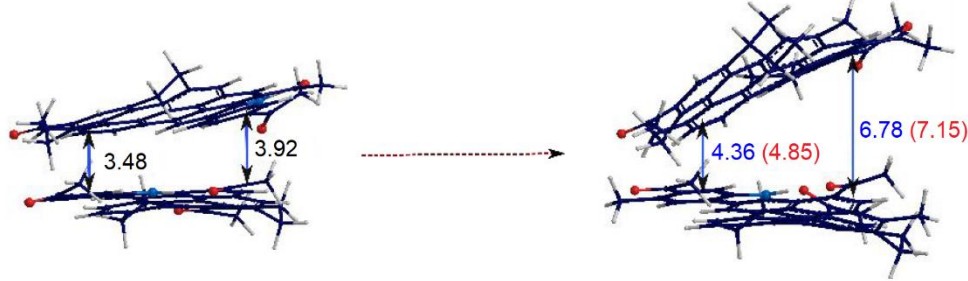

**Figure 11.** Intermolecular gaps between the aromatic rings in zones A/B and C for an oxidized dimer before (**left**) and after (**right**) rejuvenation. Blue and red numbers indicate the distances (in Å) achieved with two different rejuvenators (Reprinted with permission from [161], Copyright 2019, American Chemical Society).

*4.3. Laboratory Rejuvenating Assessment Methods: General Results*

As mentioned in the previous chapters, one way to evaluate the aging and rejuvenation of bitumen is to monitor the progress of the four SARA fractions. With oxidative aging, aromatics and resins are converted into asphaltenes leading to an increase in the contacts between the micelles in the colloidal system, increasing bitumen stiffness. The application of rejuvenators should increase the maltene content of the aged binder and consequently restore the proportions of the system.

In this regard, Zadshir et al. [172] tested an aged bitumen (RTFOT and 2 PAV) rejuvenated using three additives of different natures: the first additive was based on organic oil from pig manure (BB), the second derived from vegetable oils (VB), and the third was hydrocarbon-based (PB). The chemical composition of the biological additive (BB) shows that it is characterized by a high content of asphaltenes and resins and a low content of saturates and aromatics (Figure 12). The high content of resins stabilizes the asphaltenes and the relative micelles of the colloidal system, but the addition of further asphaltenes is typically discouraged in favor of a higher content of aromatics. Despite the structure of bio-based asphaltenes being different from that of hydrocarbon-based asphaltenes [173], the lack of aromatics and abundance of asphaltenes do not tend to rebalance the SARA composition. Conversely, the vegetable oil (VB) additive shows a great content of resins and aromatics, with low percentages of saturated and asphaltenes. The high amount of aromatic components can balance the maltenic fraction lost during the aging process, causing an increase in the colloidal stability of the aged binder. Lastly, the hydrocarbon-based additive (PB) is characterized by a large percentage of aromatics and a lower percentage of asphaltenes in comparison to the biological binder (BB).

The evolution of the percentages of the SARA fractions using these rejuvenators can also be analyzed by monitoring the Colloidal Index (CI). In particular:

- Adding the biological rejuvenator (BB) (10% by weight) causes a decrease in the CI from 0.61 of the aged binder to 0.51 (bringing it back to values similar to that of virgin bitumen). However, by increasing the percentage of the same additive up to 30%, the index increases to the unit value (exceeding even that of the aged binder without additives);
- Adding the vegetable-oil-based additive (VB), a progressive decrease in the index by increasing the percentage of rejuvenator occurs. This means that the stability of the binder is increased as the additive dosage increases. With an additive content of 30%, a CI lower than that of virgin bitumen is reached.

- Adding the hydrocarbon-based rejuvenator (PB), the percentage of asphaltenes decreases and this leads to a slight decrease in the CI. Increasing the percentage of additive from 10% to 30%, there is not a further decrease.

From the comparison between the three rejuvenators, it is clear that the content of resins and aromatics brought by the additives plays a fundamental role in the process. Resins are more effective than aromatics because of their high polarity and ability to disperse the asphaltenes micelles in the maltenic phase and consequently stabilize the colloidal system.

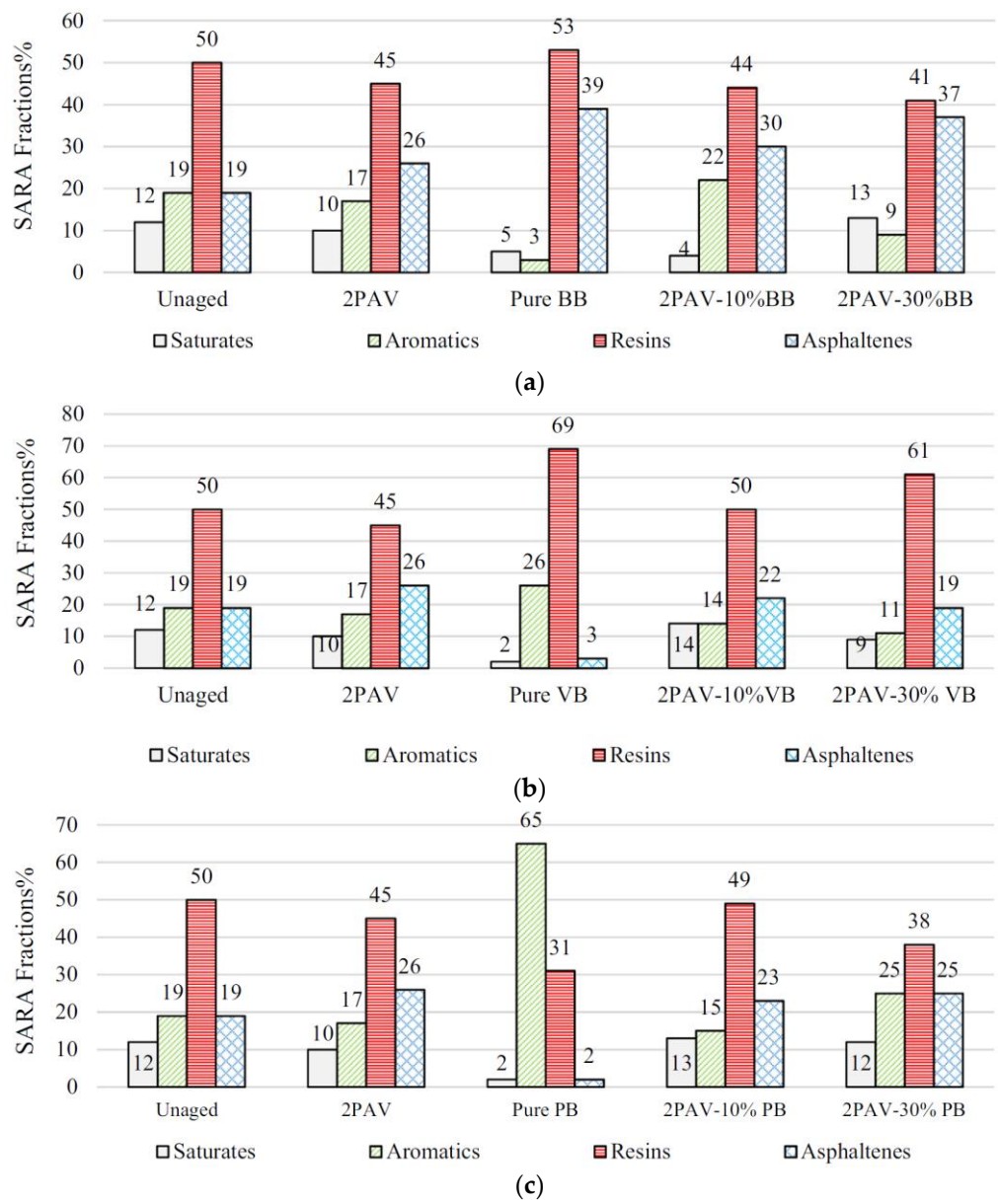

**Figure 12.** SARA fractions of virgin binder, aged binder, additives, and rejuvenated binder: (**a**) biological additive, (**b**) vegetable oil, (**c**) hydrocarbon-based additive (reprinted from Reference [172] with the permission of Elsevier).

A very important aspect to consider in the rejuvenating process is the amount of additive that has to be introduced into the aged bitumen to ensure the greatest benefit to the process. The results of CI help to find an optimal rejuvenator content for each type of additive. However, it has to be considered that for some products, as in the case of the

biological additive, due to the different nature of the asphaltene molecules that compose it, the CI may increase with increasing rejuvenator dosage, but an improvement in physical and rheological behavior of the bitumen is still achieved.

The same study [172] also investigated the molecular size distribution of the rejuvenated bitumens through GPC. The results show that:

- the large molecules (LMS) increase from 83% for the non-aged binder to 87% for the aged binder, at the expense of the percentage of medium-sized molecules (MMS), which is reduced by 14% to 10%. The increase in LMS is a consequence of the increase in the number of asphaltenes in the system and their agglomeration.
- the addition of a rejuvenator tends to decrease the percentage of larger LMS molecules by increasing the presence of medium-sized molecules (MMS).

Further conclusions, related to the GPC test, were also obtained by Cao et al. [174], who tested the effects of waste cooking oil as a rejuvenator for aged bitumen. The main conclusions obtained in this study are the following:

- The aged binder gets higher $M_w$ and $M_n$ values than the virgin one, denoting the formation of larger molecules in the binder during the aging process. Compared to the virgin bitumen, the higher poly-dispersion of the aged binder indicates that there is a greater distribution of molecular weights.
- Adding waste cooking oil with different dosages, there is no chemical reaction between the additive and the aged binder. The decrease in $M_w$ and the poly-dispersion is due to a physical dilution.

Figure 13 shows the curves obtained by the GPC of the virgin, aged, and subsequently rejuvenated bitumen with different dosages of additive.

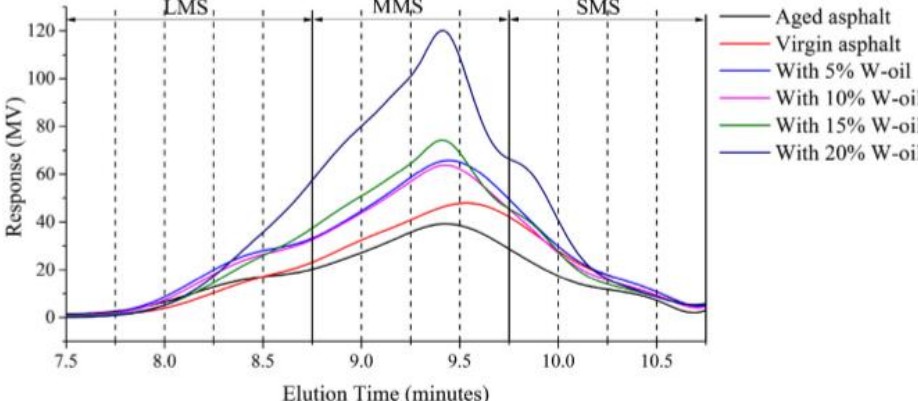

**Figure 13.** GPC curves of the regenerated binder with different dosages of waste cooking oil (reprinted from Reference [174] with the permission of Elsevier).

Compared to the virgin binder (red line), the aged one (black line) has a higher percentage of LMS compared to a lower small and medium-size molecules. Adding the waste cooking oil, the LMS tends to decrease; the typical LMS concentration of the virgin binder can be recovered by adding 20% additive. Increasing the percentage of waste cooking oil, MMS greatly increase and SMS decrease. However, the percentage of MMS and SMS cannot be brought back to the original values of the virgin binder. Therefore, it can be concluded that by adding waste cooking oil, the distribution of the molecular size cannot be totally restored.

FTIR analysis can be used to investigate the evolution of the functional groups present in an aged binder when it is rejuvenated with additives. Referring to the study made by Cao et al. [174], hereafter, a picture showing the FTIR spectra of the waste cooking oil, the virgin binder, and the rejuvenated one is reported (Figure 14). Figure 15 shows the FTIR spectra of the functional groups S=O, C=O, and C-C in more detail. Additive and bitumen do not reach together during the rejuvenating process. In fact, the rejuvenated

binder spectra are a union of those of the aged binder and the rejuvenator, without any new groups [155,175]. In terms of carbonyl index ($I_{C=O}$) and sulphoxide index ($I_{S=O}$), both increased with aging and were decreased by adding rejuvenators. This effect is related to the physical dilution of the binder.

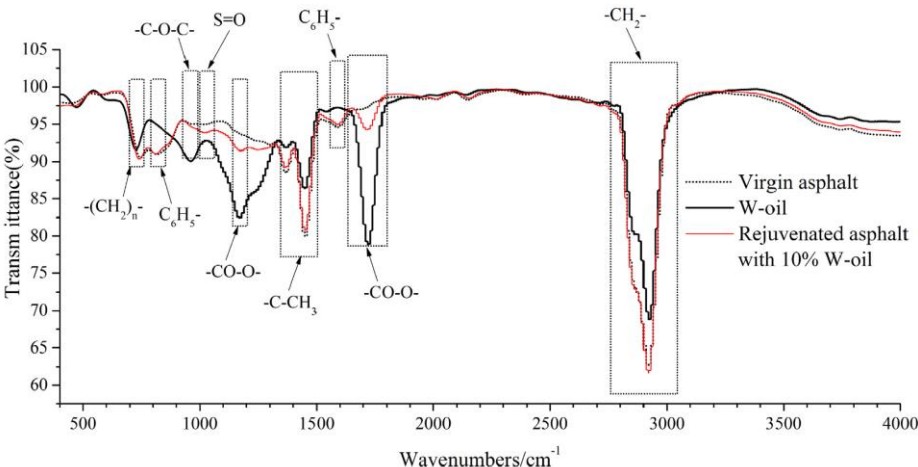

**Figure 14.** FTIR spectra of WCO (W-oil), virgin, and rejuvenated bitumen (reprinted from Reference [174] with the permission of Elsevier).

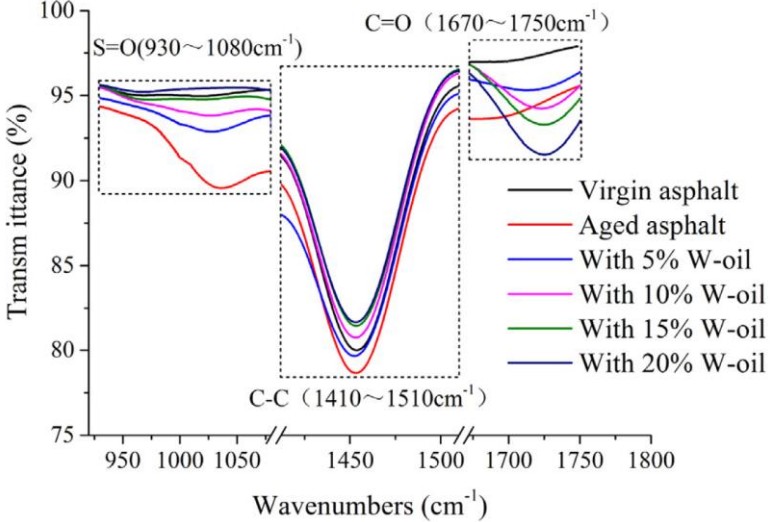

**Figure 15.** FTIR spectra of S=O, C=O, and C-C (reprinted from Reference [174] with the permission of Elsevier).

A study conducted by Noor et al. [92] focused on the analysis of the functional groups via FTIR when a biological or aromatic rejuvenator is added to a bituminous binder. Qualitative analyses were conducted by studying the spectra of the virgin bituminous binder, the biological additive, and the aromatic additive, in order to identify the representative functional groups. The functional groups identified in the bitumen are the same as those identified in the aromatic rejuvenator (both derived from petroleum). When the aromatic additive is added into the bitumen, there are no differences in the absorbance spectrum, as the functional groups are the same. On the other hand, when the biological rejuvenator is added, the resulting spectrum shows two distinct peaks in correspondence of the functional groups C=O (1744 cm$^{-1}$) and C-O (1162 cm$^{-1}$). Therefore, from the quantitative point of view, these bands can be a good reference for the identification of a specific biological rejuvenator in the bituminous binder.

Recently, Menapace et al. [176] investigated the AFM images when adding tall oil to two different types of aged bitumen:

- TOAS: blend of virgin and aged bitumen extracted from Recycled Asphalt Shingles (RAS) from re-roofing or roof removal projects;
- MWAS: blend of virgin and aged bitumen extracted from RAS from the excess material obtained during the shingles' production.

Regarding the TOAS-type bitumen, after long-term aging, there is a significant increase in roughness and a decrease in the phase contrast of the colloidal structure (Figure 16). Bee structures appear in greater numbers but less defined than they were in virgin bitumen. The reduced size of the bee-shaped structures that have formed with aging indicates a relatively lower molecular mobility than for the virgin binder. Adding tall oil, there is a reduction of the dispersed domains and an increase of the matrix area. At the same time, a slight decrease in roughness and phase contrast are noted, and the bee structures are no longer visible. After re-aging (second PAV) of the rejuvenated binder, the surface of the matrix decreases, the roughness increases, the phase-contrast decreases, and some bee-shaped structures reappear. It is interesting to understand how the bee-shaped structures come out again after the PAV re-aging, while they are not present in the rejuvenated blend. It is possible to hypothesize that the species used for the construction of the bee domains require a certain molecule agglomeration to create bee-shaped structures. The rejuvenator lowers the degree of association, thus breaks the bee-shaped structures originally present. Aging, on the other hand, helps the molecules to cluster again and reform the bee-shaped structures.

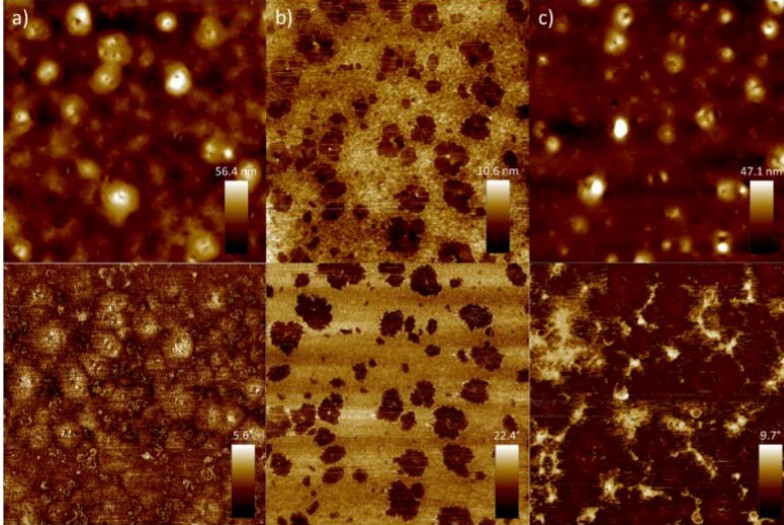

**Figure 16.** Height (**above**) and phase (**below**) images of aged TOAS binder (**a**), regenerated with tall oil (**b**) and re-aged (**c**) (reprinted from Reference [176] with the permission of Elsevier).

In the MWAS bitumen, the long-term aging seems to form rippled structures that are slightly different from the typical bees (Figure 17). When the rejuvenator is used, the dispersed domains degenerate on the binder surface as if they were liquid, to form a single interconnected phase. The rippled structures are still present, and the overall surface roughness slightly increases, while the phase-contrast shows a small decrease. After a second long-term aging process, numerous bee-shaped structures appear on the surface (c). The edges of these structures are not as smooth as in the original ones, but show a more irregular conformation. This indicates that the molecules can hardly move and reach colloidal stability.

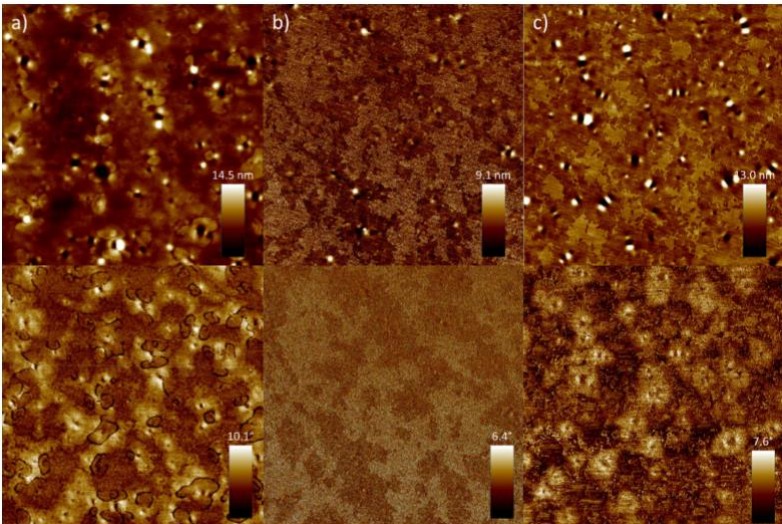

**Figure 17.** Height (**above**) and phase (**below**) images of the MWAS aged binder (**a**), regenerated with tall oil (**b**) and re-aged (**c**) (reprinted from Reference [176] with the permission of Elsevier).

In conclusion, the rejuvenator (tall oil), through the separation of the agglomerates formed by aging, allows the attenuation or even elimination of the bee-shaped structures, which, however, reappear when the binder ages again.

More recently, Ganter et al. [177] tried to evaluate the evolution of the bee structures during the aging and rejuvenating steps. For this purpose, three different rejuvenators, coming from bio (R1, R2) and oil (R3) sources were used. A PmB 25/55-55 bitumen was first short- and long-term aged (RTFOT + PAV) and then rejuvenated with the three additives listed before. They highlighted that the virgin binder displays clear bee-like structures and with increasing aging, the number of bees rises. The interesting aspect is that the addition of the three different rejuvenators causes a very different change in the surface morphology of the bitumen: size and quantities of the bee-like structure differ completely as a function of additive type and origin.

## 5. Aging of a Rejuvenated Bitumen (Re-Aging)

The growing use of RAP in the road sector is leading to new pavement mixtures composed not only of virgin materials, but also of old recycled bitumen and rejuvenators. Therefore, it is very important to understand the changes that occur in the properties of an HRMA when it is subjected to aging during its service life (which indeed is a "second aging" or "re-aging" for the RAP material). Hereafter, a literary review regarding the re-aging changes in the chemical and rheological properties of the recycled bitumen is provided.

Generally, the second aging in a rejuvenated binder is less harmful with respect to the aging of a virgin bitumen. Based on the studies carried out by Mazzoni et al. [178] and Bocci et al. [179], it is reasonable to expect that HMA including a high amount of RAP and a rejuvenator suffers fewer aging phenomena and can be less stiff than an alike mixture with no RAP.

From the chemical point of view, in a recent study, Ingrassia et al. [2] focused on evaluating the possibility of recycling binders that were already rejuvenated. Two virgin binders (one ordinary bitumen and a bio-binder) and two aged binders (one recovered from RAP and one "Bio-RAP" binder produced in the laboratory) were combined to reproduce the RAP hot recycling process. The chemical properties before and after aging were investigated using FTIR analysis through the evaluation of the $I_{C=O}$ and $I_{S=O}$ indices and an additional parameter, named chemical aging index $AI_{FTIR}$, and calculated with the following equation:

$$AI_{FTIR} \frac{(Ico + Iso)_{aged}}{(Ico + Iso)_{unaged}}$$

Basically, the analysis of $AI_{FTIR}$ shows that the susceptibility to aging of recycled blends is significantly lower than that of the reference virgin binder. In fact, recycled bituminous blends (already containing a certain amount of oxidized binder) are less susceptible to further long-term aging.

Moreover, Sa-da-Costa et al. [180] tried to analyze, from the chemical point of view, the effects of aging on a rejuvenated bituminous binder. Several types of bitumen were analyzed in this study. As expected, an increase in the content of asphaltenes was found, confirming the oxidative transformations that contribute to increasing the polarity of the bitumen components in terms of SARA fraction. Furthermore, the oxidation of the bitumen involves a general increase in the carbonyl functional groups (C=O). Although precise correlations have not been found out, it can be stated that the chemical properties of an aged rejuvenated bitumen (second aging) are very close to that of an aged virgin binder (primary aging).

The second aging is of fundamental importance to understand the condition of the HRMA at the end of its second useful life. Continuous recycling of RAP would allow an increase in circularity related to the road construction field. For this reason, in the next future, more studies regarding this aspect need to be carried out.

## 6. Conclusions

In the sector of road pavement engineering, the easiest way to promote the circular economy is to encourage the use of RAP. One of the main issues regarding the use of this recycled material is the aged bitumen contained in it. Since the aging and rejuvenating process of bituminous pavements can significantly affect their durability and service life, it also is fundamental to achieve a complete knowledge of these aspects from a chemical point of view.

As explained in detail in the previous sections, bitumen aging involves different phenomena: loss of volatiles, oxidation, and physical and steric hardening. These result in the unbalancing of the SARA fractions and the formation of large molecular agglomerates, which in turn affect the mobility of the fractions in the colloidal system and determine the binder embrittlement. In order to restore the properties that bitumen loses with aging, rejuvenators can be used. A good rejuvenator can rebalance the SARA components and, despite oxidation being mainly irreversible, it can disrupt the asphaltene clusters and re-establish the proper molecular mobility.

In the present review, different chemical analyses were described as tools to investigate bitumen aging and rejuvenation processes. Table 6 provides an overview of the discussed approaches. All the techniques showed promising results but, due to the limitations of each method, the contemporary use of different approaches is highly recommended to have a precise and clear overview of how bitumen ages and rejuvenates. Moreover, the definition of a multi-testing protocol for the characterization of the effects of aging and rejuvenation at a chemical level can represent the basis for future application on more complex binders (polymer-modified bitumens, bitumens including extenders or nanoparticles).

**Table 6.** Advantages and limitations of the different techniques for the study of bitumen aging and rejuvenation.

| Technique | | Investigation of Aging | Investigation of Rejuvenation |
|---|---|---|---|
| AFM | Advantages | The bitumen chemo-morphological degradation with aging can be studied using AFM. | The rejuvenation process has been proven to influence the surface morphology of the aged bitumen, specifically the bee structures. |
| | Limitations | Since the approach is very recent, there is still a gap of knowledge on associating the AFM phase evolution with other chemical and mechanical properties. | The experimental approach is still at an early stage, and further research is necessary to deeply understand how to exploit this powerful tool. |
| FTIR | Advantages | It allows determining the severity of aging through the change of specific bands (particularly sulfoxide and carbonyl). | The presence of the rejuvenator in an HMA can be detected by comparing the spectra of the pure additive and the recovered bitumen. |
| | Limitations | It does not discriminate what happens to the bitumen colloidal system, but mainly focuses on the oxidation effects. | Since most additives have peculiar bands in correspondence to the sulfoxide and carbonyl bands, it is difficult to quantify the rejuvenator content in the recovered bitumen. |
| TLC-FID | Advantages | The evolution of the SARA fractions with aging allows estimating the severity of the phenomena. | It allows understanding the efficacy of the rejuvenation process in restoring the SARA proportioning. |
| | Limitations | There is not a clear correlation between the SARA proportions and the bitumen mechanical behavior. | Again, the main issue with this technique is related to the poor association between SARA proportions after rejuvenation and the effective improvement of the aged bitumen rheological properties. |
| HP-GPC | Advantages | It allows quantifying the effects of SARA fraction shifting and the agglomeration of the asphaltenes due to aging. | The analysis of the molecular weight distribution allows one to understand if a rejuvenator can really detach the asphaltene clusters or only has a dilution effect. |
| | Limitations | There is still uncertainty on how the different aging phenomena (oxidation, loss of volatiles, etc.) influence the molecular weight distribution. | It could be used to estimate the degree of blending between aged and virgin bitumen with/without rejuvenators, but no precise procedures have been defined yet. |

## 7. Recommendations for Future Research

The state of the art provided in the present paper highlighted that, despite extensive research having been moved to innovative and improved binders, there are still important aspects about neat bitumen that deserve to be studied to fully understand the complex chemistry of this material and somehow predict the behavior of the bituminous mixtures when in service. In light of the themes discussed in the previous sections, future works should be addressed to:

1. Combine the results of the chemical tests at the binder scale with the results of the mechanical and rheological tests at both the binder and mixture scale. Understanding what happens to the bitumen from the chemo-morphological point of view is fundamental, but should be correlated to the corresponding effects on the material performance in order to have the research focused on the practical outcomes. To this goal, for instance, the IFSTTAR research team recently identified a relationship between the bitumen molecular weight distribution and the phase angle of the complex modulus, and proposed a tool, the δ-method, to determine the molecular weight distribution from rheological tests [181]. Within the RILEM TC 264-RAP, scientists are trying to find links between the mechanical characteristics of HRMA and the FTIR spectrum of the extracted bitumen, while also aiming to estimate the presence or even the content of a rejuvenator in a mix from the FTIR binder spectrum.

2. Evaluate new solutions to hinder, restrict, or slow down the bitumen's aging. Several investigations showed that the use of a straight-run bitumen, instead of a visbreaker

one, can reduce the aging susceptibility of an HMA in both the short and long term [29,30,182]. However, poor attention is paid by road authorities, boards for standardization, and HMA manufacturers on the bitumen's origin and production process. In a similar scope, further research should also be focused on additives with antioxidant effect, that is, with the ability to reduce the bitumen's propensity to oxidation. Some products are currently available on the market with the declared effect of hindering bitumen oxidation, but scientific studies are required to deeply understand their behavior at both the chemical and mechanical levels.

3. Understand the interaction between old and fresh bitumen in hot recycling. The topics of RAP bitumen degree of activation (DoA) and RAP/virgin bitumen degree of blending (DoB) are actually among the most studied worldwide [183–193]. However, because of the huge complexity of the problem (which is influenced by many factors such as RAP bitumen content, nature, and aging state; HMA production process; type, dosage, and method of addition of the rejuvenators; hauling, paving, and compaction procedure, etc.), univocal protocols to classify different RAP materials according to the DoA or to estimate the DoB during pavement construction have not been defined.

4. Identify a method to allow precise quality controls on HRMA. This objective, which is maybe utopian, is one of the most crucial. Technical specifications currently provide controls on the HMA mechanical performance to limit the amount of RAP in the mix and encourage the use of rejuvenators. However, a solution to estimate how much RAP and how much rejuvenator have been included in a mix should be found, possibly including a series of physical, chemical, microscopic, and rheological analyses on the raw materials (RAP and its components, rejuvenator, virgin bitumen), the laboratory, and the plant-produced mixtures preliminarily to full road construction.

**Author Contributions:** Writing—original draft preparation, E.P.; writing—review and editing, E.B. Both authors have read and agreed to the published version of the manuscript.

**Funding:** This research received no external funding.

**Institutional Review Board Statement:** Not applicable.

**Informed Consent Statement:** Not applicable.

**Data Availability Statement:** Not applicable.

**Conflicts of Interest:** The authors declare no conflict of interest.

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
