# Peer review of "A Review on Bitumen Aging and Rejuvenation Chemistry: Processes, Materials and Analyses"

_sustainability, doi:10.3390/su13126523_

Round 1

Reviewer 1 Report

This paper shows a good state-of-art on bitumen aging. There are some issues that need to address:

- I believe this review would benefit from a table that could compare and provide an overview of the discussed approaches. The table should include the advantages and limitations of each approach. In table 4, it's better to add the contents of Additives.

- The language of the paper needs to be improved, as such, it is really difficult to read...

- Introduction is written simply, most recent research and innovation bitumen aging and rejuvenator composites performances should be reviewed to show the gap of knowledge. The introduction should be extended with recent research papers. The introduction should be rewritten to show the highlights and novelty of the work.

- section of drawbacks and future could be increased quality of manuscript.

- Similar reviews have been published recently. It is recommended to add a statement to clearly separate the current work from these similar references and also define the review period (e.g. last five years). Also, prepare statistical data (such as the number of documents, document per country) about you used references by created databank such as Scopus, Google scholar, and web of science.

Authors can use below references and related works in the field:

-  "A review on Bitumen Rejuvenation: Mechanisms, materials, methods and perspectives." Applied Sciences 9, no. 20 (2019): 4316.

- "Effect of Fumed Silica Nanoparticles on Ultraviolet Aging Resistance of Bitumen." Nanomaterials 11, no. 2 (2021): 454.

-  "Characterization of bitumen modified with pyrolytic carbon black from scrap tires." Sustainability 11, no. 6 (2019): 1631.

- "Ultraviolet aging study on bitumen modified by a composite of clay and fumed silica nanoparticles." Scientific Reports 10, no. 1 (2020): 1-17.

- "Short-term aging effect on properties of sustainable pavement asphalts modified by waste rubber and diatomite." Sustainability 9, no. 6 (2017): 996.

Reviewer 2 Report

Detailed review of bitumen aging and rejuvenation is presented in this article with extensive explanations of chemical processes. However, article is too long and some parts are tedious for reading. In order to achieve more readable article, I suggest following corrections:

Section 2.4. should be reduced. Investigation of bitumen chemistry is not absolutely necessary to understand bitumen aging from civil engineering point of view. The same goes for section on laboratory aging methods description (lines 399-413). Section 4.2. Rejuvenating mechanisms should be also omitted or rewritten since it is basically presenting research theory of another researcher.  

Typo: line 321, While should be while.

Typo: line 392, word “test” is repeated twice in “rolling thin film test oven test”.  

Round 2

Reviewer 1 Report

The comments of my first report have been addressed by the authors.